# The Use of SMAV Model for Computing Fragility Curves

**Vitantonio Vacca [1], Giuseppe Occhipinti [1,*], Federico Mori [1] and Daniele Spina [2]**

1  Institute of Environmental Geology and Geoengineering, Italian National Research Council (CNR), 00010 Rome, Italy
2  Department of Civil Protection-Seismic Risk Office, 00189 Rome, Italy
*  Correspondence: giuseppe.occhipinti@igag.cnr.it

**Abstract:** Among the buildings with strategic role for civil protection purposes part of them were built before the introduction of modern seismic codes. Aiming to guarantee their operability in case of seismic events, they have to be assessed and, hopefully, seismically upgraded. This aspect arises the necessity of reliable low computational demanding numerical models and probabilistic strategies that can be easily adopted in case of the assessment of conspicuous building stock. Part of the authors proposed and validated, in previous works, a low-demanding seismic model calibrated on ambient vibrations (SMAV) that is engaged in this paper for calculating fragility curves. On the base of dynamic and geometric properties, the SMAV model simulates the dynamic response of buildings. The herein proposed procedure adopts the SMAV model and takes into account the direction of seismic input in a probabilistic manner. The use of the ASI scale as an Intensity Measure parameter is discussed and the results are compared to those obtained with others IM scales. Although SMAV implements an equivalent linear procedure the results are compared with those obtained by more sophisticated FEM models and nonlinear dynamic analyses with a satisfactory agreement. Finally, two real case studies are presented. The fragility curves for both the case studies are calculated and discussed. The results encourage the authors to the use of fragility curves based on SMAV model. Additionally, the fragility curves can be easily implemented in seismic risk assessment procedures at urban or territorial scale and, consequently, the proposed procedure may represent a key aspect in the reduction of seismic risk and rationalization of investments for the seismic upgrading of strategic buildings.

**Keywords:** SMAV; fragility curves; existing buildings; civil protection; concrete frame structures; ASI; PGA

## 1. Introduction

Recent advancements in computational engineering enhanced the simulation accuracy by means of high-fidelity models of entire buildings [1] leading to extremely accurate numerical simulations. On the other hand, the amount of existing structures that were not designed to withstand earthquakes requires a critical focus towards low-cost reliable strategies for damage assessment at large or urban scale [2,3] for expected seismic events. Aiming to provide general results, based on a probabilistic approach, it is common to provide the probability of damage occurrence on a structure in terms of a measure of the seismic intensity, usually denoted as fragility curves [4–6]. Efforts have been already carried out in this direction in terms of simplified models [7,8] that can be related to the seismic vulnerability at large [9] or building [10] scale by some of the present authors. In particular, two of the authors proposed the SMAV (Seismic Model from Ambient Vibrations) strategy [7,8,11] that is a mathematical model for predicting the seismic response of an existing building subjected to earthquake that produces no severe damage. The model is based on the identification of modal parameters from ambient vibration together with few information about its geometry and constructive typology. According to the SMAV approach the non-linear behaviour, which typically occurs during even non-destructive earthquakes,

is considered by assigning appropriate laws of variation to the natural frequencies as the deformation increases.

In this paper, a SMAV model-based framework to evaluate fragility curves, applied to multi-storey buildings, is presented.

On the other hand, one of the authors proposed a linear elastic beam-like model in previous works [12–14] that is another way to perform low computational burden analysis on large scale. As this author claimed in a previous work, "the use of reliable numerical models based on a reduced number of structural or dynamic information", such as the equivalent beam-like model or SMAV model that is engaged in this work, "represents a powerful tool with economic and decision-making advantages" [12].

This study exploits and implements the use of the SMAV model that plays a key role in an ongoing research project in collaboration with the Italian National Research Council and the seismic risk office of the Department of Civil Protection within a PON Governance 2014–2020 project on the reduction of seismic, volcanic and hydrogeological risk for civil protection purposes [15]. That research aims to propose efficient seismic assessment strategies to be applied to existing buildings with emphasis to structures possessing a strategic role.

In this proposal, in order to estimate the nonlinear response based on equivalent-elastic analyses, the interstorey drift ratio has been adopted as Engineering Damage Parameter (EDP). As matter of fact, it is worth clarifying that the latter parameter has been chosen among several other possible EDPs [16] since it appears to be the most reliable a-priori estimation to predict damaged configurations based on a linear structural response. Other EDPs for RC structures depend on the knowledge of the structural geometries and details of the entire building hence the adoption of the interstorey drift ratios as EDP appears to be adequate also in view of extensive and economically affordable numerical calculations. The proposed framework, starting from the modal properties obtained by dynamic identification campaigns, firstly aims at formulating a suitable, sufficiently accurate, SMAV model and, successively, constructing fragility curves through dynamic analyses. The herein presented fragility curves are obtained by analysing the interstorey drift ratios obtained mainly for Peak Ground Acceleration (PGA) and Acceleration Spectrum Intensity (ASI) values that define the Intensity Measure (IM) scales of the considered seismic events. In particular, several real accelerograms of earthquakes that stroke the Italian peninsula have been selected to develop the fragility curves [17].

Aspiring to validate the use of SMAV method, the fragility curves of a multi-storey building benchmark have been compared with those obtained by means of an accurate nonlinear FEM model [18].

In view of the adoption of the fragility curves for Civil Protection purposes the method is applied to those structures that belong to Strategic Buildings, in the Civil Protection layout, that have to be affected only by negligible damages in case of earthquake [3]. In order to mitigate the seismic risk at territorial scale by reducing the seismic vulnerability of those structures low computationally demanding strategies have to be established.

## 2. The SMAV Model

Aiming to calculate model-based fragility curves, high-fidelity [1] or simplified [12] model of the existing buildings may be implemented. In previous research, some of the present authors developed a simplified Seismic Model from Ambient Vibration (SMAV) [8,11] that has been implemented in the methodology presented in the following sections. The SMAV model allows to reproduce the seismic behaviour of the structure on the base of ambient vibration measurements. The model, that guarantees low computation burden and reliability of results as discussed in previous works [8], allows to perform several analyses as in case of defining fragility curves. SMAV model is implemented in VaSCO-smav software [19].

From a theorical point of view, the starting point of SMAV is the Multi Rigid Polygons model [20] employed for estimating the seismic mass associated to each mode shape and

for extrapolating the dynamic response of the building even in points not directly measured during the ambient tests. To this end each floor of the building is ideally divided into several polygons, which are assumed to have a rigid behavior in their horizontal plan, and, consequently, ambient vibrations are measured using two biaxial accelerometers on each of these polygons. The mode shapes extracted from the recorded signals, via operational modal analysis, are then expressed in terms of rigid Degree of Freedom (DOFs) of each polygon (two translations and one rotation) with respect to their geometric center of gravity, in which the inertial properties of the building are also concentrated. In this way it is possible to compute the mass matrix of the building and so to obtain the seismic mass associated with each mode shape. Once experimental natural frequencies, mode shapes and modal masses have been identified (for damping ratio the conventional values of 0.05 is considered), it is possible to build a linear model that provides the seismic response of the building by mode superposition, first in the rigid DOFs of the polygons and then, through a simple linear transformation, in the DOFs of points along the perimeter of each floor of the structure.

Moreover, SMAV is able to take into account the non-linear seismic behaviour of the building thanks to the so-called "Frequency Shift Curves" [7,8] (FSC) developed for some specific structural typology, which allow the roof drift of the building during the earthquake to be correlated with the decrement (shifting) of its the apparent natural frequencies. FSCs are used within an iterative procedure in which the results obtained in step ith are used to calculate the value of the natural frequencies adopted in step ith + 1, until the achievement of pre-established conditions of convergence. It is important to underline that the probabilistic nature the "Frequency Shift Curves", linked to the uncertainty on the mechanical and geometric parameters of the structure, is not considered in the present work. In the following, in fact, the values of the maximum inter-story drift will be obtained using the "Frequency Shift Curve" corresponding to a frequency shift that, for an assigned seismic input, has a probability of exceeding equal to 0.5. This curve will be considered as the one that characterizes the deterministic behavior of the building.

Finally, it must be emphasized that SMAV is based on the hypothesis that, while frequencies may vary due to the nonlinear behaviour or damage, the mode shapes are considered invariant. For this reason, the model is strictly applicable up to moderate damage level.

## 3. Methodology

This section exposes a novel procedure for calculating fragility curves of existing buildings on the base of ambient vibration test and a simplified numerical model. According to other model-based fragility curve procedures the proposed framework is divided into three phases: (i) Phase 1: definition of the inputs, (ii) Phase 2: definition of the numerical model; (iii) Phase 3: analysis of the results.

It is worth noting that an innovative aspect of the method is the use of a simplified numerical model SMAV, above described. Since this model accounts for the spatial structural behaviour of the building, pairs response spectra, calculated from natural accelerograms, are applied spatially in two directions for each analysis. In the procedure, the Multiple Stripes Analysis (MSA) [21] is engaged for developing fragility curves.

Attempting to investigate the capability of the numerical model to define reliable fragility curves and their dependency on the Intensity Measure scale, several analyses were performed and the results were compared.

One of the limits of the fragility curve is to express the probability of damage as a function of a unique parameter defined intensity measure (IM). Aiming to describe the intensity of the input several researchers proposed different parameters [22–24]. The IM parameters are computed in two spatial directions. Therefore, in order build a two-dimensional fragility curves, it is necessary to condense the vector of physical intensity

parameter in both directions [$p_x$ $p_y$] into a single scalar value $p$ given by the following geometric mean:

$$p = \sqrt{p_x \cdot p_y} \tag{1}$$

In the present research, response spectra obtained from natural and synthetic accelerograms have been considered. Each pair of response spectra was applied to SMAV, and the computed maximum inter-storey drift of the building was considered for estimating the damage levels that the structure exhibits under seismic forces [25–27] even in case of severe earthquakes [28].

Natural accelerograms (directed along the two orthogonal directions) of the European Strong Motion Database [17] were selected and oriented along the two principal direction of the analysed structure.

Though the Peak Ground Acceleration (PGA) scale is generally assumed as IM scale for Fragility Curves in case of existing buildings, the relationship between PGA and structural damages during earthquakes is not straightforward. As other authors already discussed [29] other IM scales may be considered if a more direct correlation with the dynamic response of the building is needed. Trying to investigate this assumption within the proposed approach, the Acceleration Spectrum Intensity ($ASI_{T_a - T_b}$) [30] IM scale is adopted and compared to the PGA scale. Acceleration Spectrum Intensity is the integration value of the Pseudo-Acceleration over a periods range ($T_a, T_b$).

Generally, fragility curves provide the probability ($P$) of exceeding the damage level (*D*) for a certain value of intensity measure (*IM*). The develop of fragility curves is based on some main assumptions:

(i)   the achievement of the damage level (*D*) corresponds to the passing of a certain threshold that is expressed in terms of Engineering Damage Parameter (EDP), the interstorey drift ratio, for instance;
(ii)  The interstorey drift ratio ($\delta$) is usually considered reliable damage index;
(iii) The probability is defined as a log-normal cumulative distribution function.

The equation of the fragility curves is reported in Equation (2).

$$P(D \mid IM) = \Phi\left(\frac{ln(IM/\theta)}{\beta}\right) \tag{2}$$

where:

$\Phi$ is the cumulative of the normal distribution;
$\theta$ is the median of $\Phi$;
$\beta$ is the standard deviation or the dispersion of $\ln(IM)$.

In this paper, $P(D \mid IM)$, is calculated using the Multiple Stripes Analysis (MSA) method [21]. MSA considers the structural behaviour of a structure only for a certain level of *IM*. At each level, the structural response is computed for an adequate number of inputs in terms of *IM* value (PGA and ASI in this research). For each *IM* level, the result is expressed as the percentage of seismic inputs that determine the exceedance of a certain damage level. Lastly, the results are fitted with the maximum likelihood method [31–33]. It seems meaningful to remember that this methodology leads to a numerically ill-conditioned estimate of the parameters $\theta$ and $\beta$ if the selected *IM* levels are close to each other or if they coincide with the peripheral sides of the fragility curve. A less distributed interval of *IM* values, approximately around the median of the fragility curve, helps to estimate correctly the real median value. On the other hand, a spread *IM* interval guarantees a satisfactory estimation of the parameter $\beta$.

## 4. The Numerical Benchmark Model

Endeavouring to validate the SMAV model to obtain reliable fragility curves, a numerical model has been developed on the base of previous works [34,35] in which the main structural and architectural characteristics of a typical reinforced concrete buildings built in

Catania (Sicily, south of Italy) before the introduction of the seismic code was defined and a case study generated.

The case study is a ten-storey RC framed residential building. The model has a rectangular plan (31.80 m × 13.80 m) and a symmetry axis oriented along the shorter dimension. The columns are placed at a centerline distance of 4.50 m in both directions. The interstorey height, except the first of 4.30 m, is 3.30 m.

A unique structural floor typology was defined. Precisely, the floor system consists in one-way RC ribbed slabs spanning orthogonally to four main frames. Figure 1 shows the plan of the generic floor and depicts the structural plan in which the ribs direction, the concrete columns orientation and the beams types are defined. The stairs slab is 15 cm thick. Since the building has been designed for gravity loads, the bending moment and shear force distributions in the beams at each floor are the same, while the column sections vary according to the floor level. A tributary area load was used for the column designs. The cross-section optimisation implied a decrement of the column cross-sections from the base floor to the top floor. As Figure 2 shows, the ground floor is signed by large openings that may determine soft-storey mechanisms. Only few peripheral panels do not present openings and, consequently, they are considered in the numerical models as discussed in the next section.

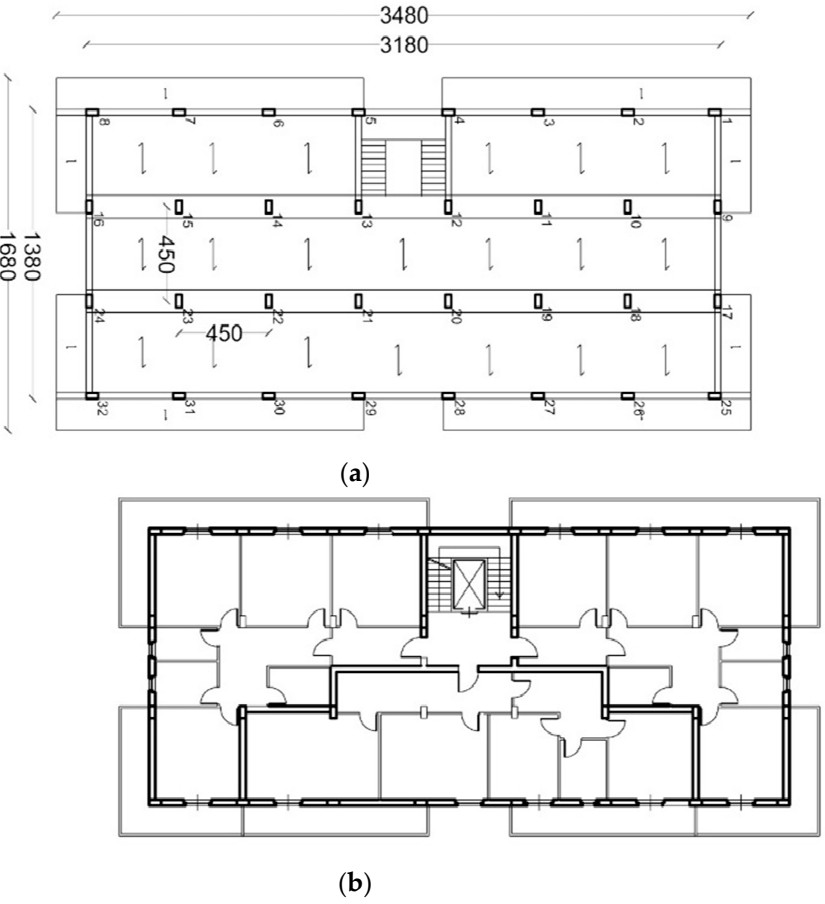

(a)

(b)

**Figure 1.** Generic architectural (**a**) and structural (**b**) plan of the case study.

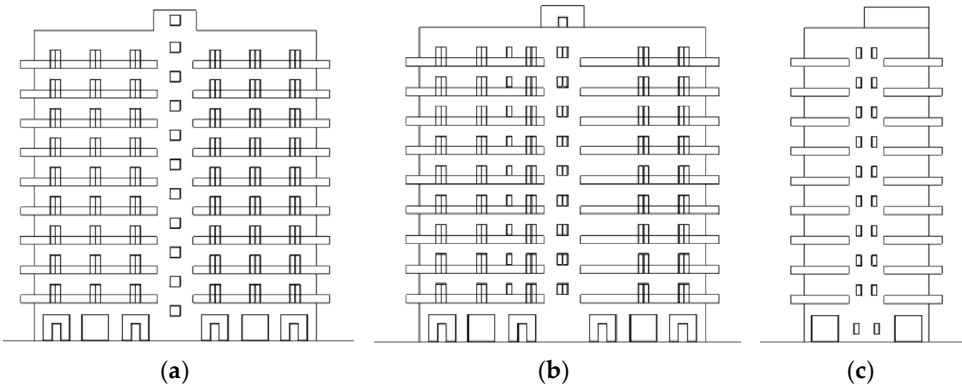

**Figure 2.** Architectural (**a**) front, (**b**) back and (**c**) lateral view.

The concrete beams have been designed for gravity loads only with line loads that represented normal condition of occupancy according to the regulation before the 1981. The reinforcement layout has been based on numerously available technical drawings and the longitudinal reinforcement have been assumed straight and shaped. Figure 3 shows the external beam reinforcement layout in the shorter direction.

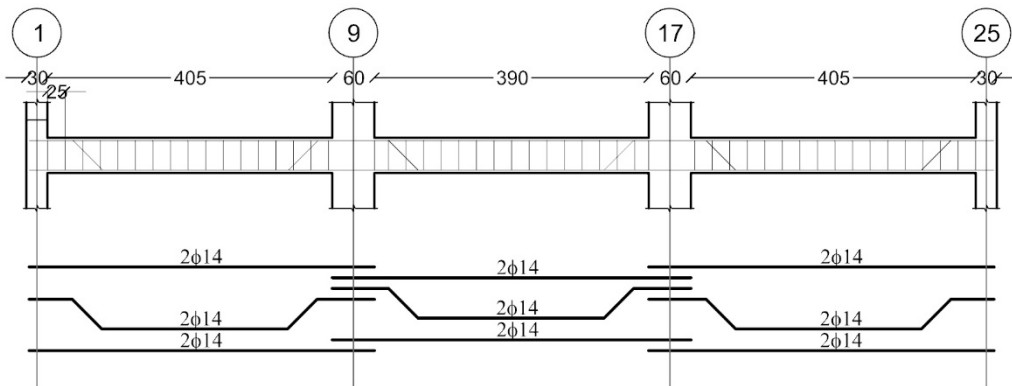

**Figure 3.** Example of reinforcement layout.

The mechanical definition of the concrete and steel materials has been based on the common practice of that time and supported by the results of structural health monitoring campaigns that were carried out on building in the south east of Sicily where Catania is placed. Tables 1 and 2 expose the main mechanical material parameters and their values.

**Table 1.** Characterization of concrete materials.

| Material Property | Cover | Core | Unit |
|---|---|---|---|
| Cylinder Compressive strength (MPa) | 20.75 | 23.25 | (MPa) |
| Young's modulus (MPa) | 27.386 | 27.386 | (MPa) |
| Strain at maximum strength | $2 \times 10^{-3}$ | $2 \times 10^{-3}$ | - |
| Tensile strength (MPa) | 1.04 | 1.04 | (MPa) |

**Table 2.** Characterization of steel materials.

| Material Property | | Unit |
|---|---|---|
| Yielding strength (MPa) | 375 | (MPa) |
| Young's modulus (MPa) | 210,000 | (MPa) |
| Strain-hardening ratio | 0.01 | - |

Two wythes of bricks with an interposed air chamber define the 30 cm thick infill panel: the thickness of the interior brick layer is 8 cm, while the exterior one is 12 cm. This

infill configuration was largely diffused and common in the south of Italy between 60's and 80's. The thicker layer (12 cm, exterior) has been taken into account only for evaluating the unreinforced masonry panel contribution.

The masonry panel characteristics are summarised in Tables 3–5 and have been based on the experimental test conducted on an infill frame subjected to cyclic horizontal loads [36].

**Table 3.** Flexural behaviour of masonry walls.

| Material Property | Value | Unit |
|---|---|---|
| Young's modulus | 1200 | (MPa) |
| Tensile strength | 0.08 | (MPa) |
| Compressive strength | 0.8 | (MPa) |

**Table 4.** Shear behaviour of masonry walls.

| Material Property | Value | Unit |
|---|---|---|
| Shear modulus | 600 | (MPa) |
| Shear strength | 0.05 | (MPa) |
| Friction angle | 0.4 | - |

**Table 5.** Sliding behaviour of masonry walls.

| Material Property | Value | Unit |
|---|---|---|
| Cohesion | 0.4 | (MPa) |
| Friction angle | 0.7 | - |

Based on the geometrical and mechanical properties of the above benchmark 3D frame building structure, a nonlinear FEM model has been implemented in the Sap2000 code [18]. The FEM accounts for the constitutive nonlinearity, P-Delta effects and spatial behaviour of the building. The 3D model (Figure 4) accounts for the effect of the non-uniform reinforcement layout of concrete beams, as already represented in Figure 3, by dividing all the beam FE elements in three parts. PMM and M plastic hinges have been applied to columns and beams, respectively. Each concrete beam can undergo plastic hinge occurrences that account for the irregularities due to the internal reinforcement layout. As reported in Figure 4b,c the infill frames are modelled by exploiting a single strut approach.

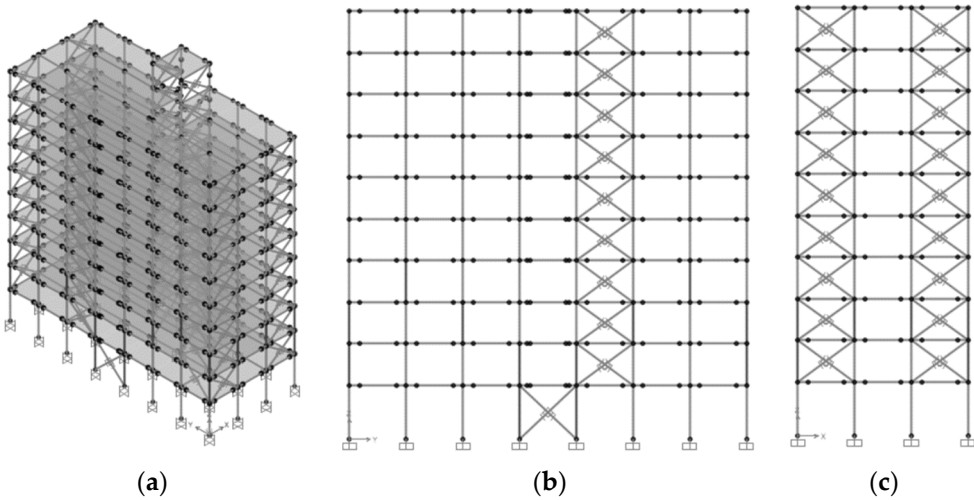

(a)　　　　(b)　　　　(c)

**Figure 4.** The numerical model in (**a**) 3D view; (**b**) longitudinal frame and (**c**) transversal frame.

Each single strut has been modelled employing one nonlinear link with a nonsymmetric constitutive law based on the Panagiotakos and Fardis's research [37], depicted in Figure 5.

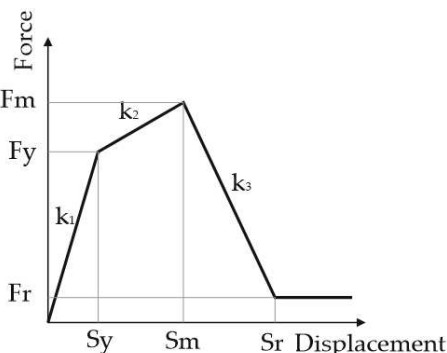

**Figure 5.** Force-Displacement relationship for the equivalent strut model proposed in [37].

Even a non-realistic [38] interaction between surrounding concrete frames and masonry panels is simulated by means of the single strut model; the latter model, in view of the massive number of dynamic analyses may represent a suitable compromise for containing the computational burden. On the other hand, for simulating as much as possible accurately the infill contribute, a staged construction method was considered in order to avoid unrealistic compressive stresses on masonry struts. Furthermore, it has to be remarked that, since the infill frames are modelled only if no openings are present, the computational model is characterised by an irregular distribution of the struts.

Preliminarily, two pushover analysis in the two main directions have been performed. The results can be adopted for evaluating the correlation between the top floor displacement and the global response. Figures 6 and 7 report the two capacity curves among x and y direction and the collapse shapes. Initially, the structure reacts with a higher stiffness due to the contribute of the infill frame in x direction, after 1–2 mm the contribute of the masonry ties stiffness becomes negligible and the structure acts with an almost linear behaviour up to 10 mm in x direction. The loss of the tensile masonry contribute is not relevant among y direction due to the presence of peripheral floor concrete beams. The average drift ratio of 3‰ is equal to 9.9 mm. Figures 6a and 7b report a local collapse of the 5th floor.

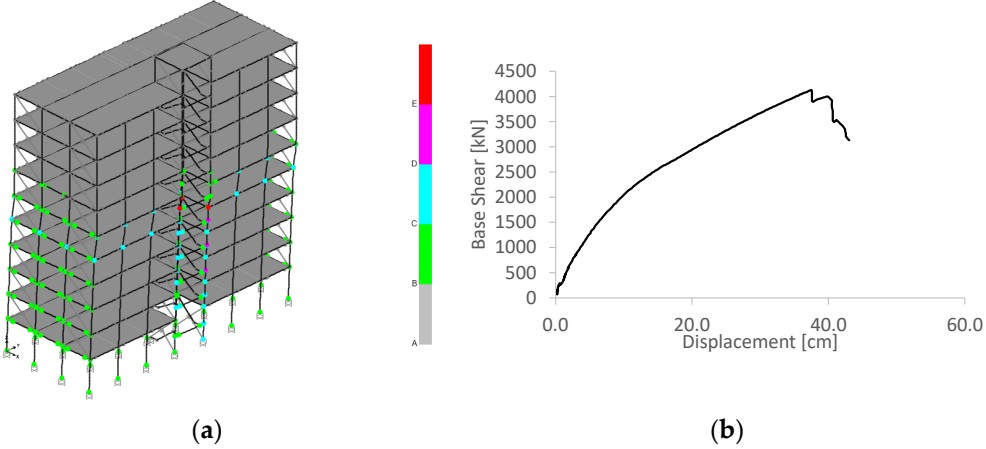

|(**a**)|(**b**)|

**Figure 6.** Pushover analysis, (**a**) deformed shape and (**b**) capacity curve in x directions.

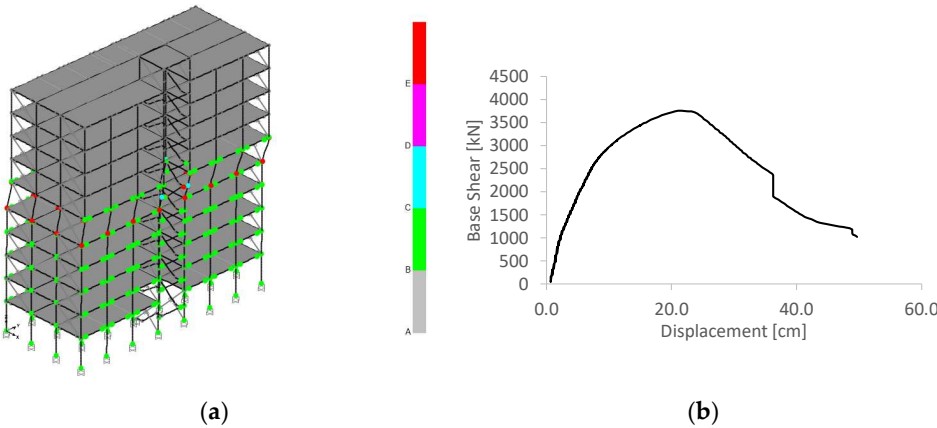

(**a**)                                        (**b**)

**Figure 7.** Pushover analysis, (**a**) deformed shape and (**b**) capacity curve in y directions.

Figure 8 reports the view of three modal shapes for which the frequencies are $f_1 = 0.695$ Hz, $f_2 = 0.721$ Hz, $f_3 = 0.915$ Hz.

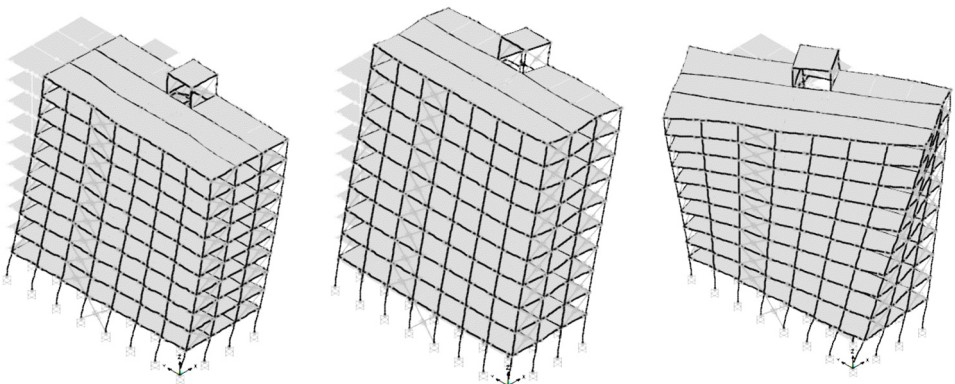

**Figure 8.** Modal shapes of the FEM model.

A significant number of independent events was used for defining site-independent fragility curves. Globally, 450 events [17] that occurred between 1972 and 2010, with magnitudes between 4.1 and 6.9 Mw on the Italian peninsula were selected. The wide range of signals employed can be considered statistically representative.

Each pair of seismic selected events was engaged for performing nonlinear dynamic analyses on the nonlinear FEM model and pseudo-elastic response spectra analyses on the SMAV model, according to the procedure described in Refs. [7,8]. This last was built using the natural frequencies and mode shapes computed with the FEM as they were experimental values extracted from ambient vibrations. Consequently, two numerical methods (FEM and SMAV) that are marked by different level of sophistication and two numerical methods with different level of computational burden have been involved. The high accurate nonlinear FEM model with the extremely reliable nonlinear dynamic analyses was compared to the simplified SMAV model that account for a less computational demanding numerical method. The present section shows how the simplified approach on the base of the SMAV model can reach satisfactory results in comparison to more detailed method. Figure 9 compares the two models in terms of maximum drift ratio ($\delta_{max}$) calculated for each event. The events are arranged in PGA and $ASI_{0.7-1.1}$ scales.

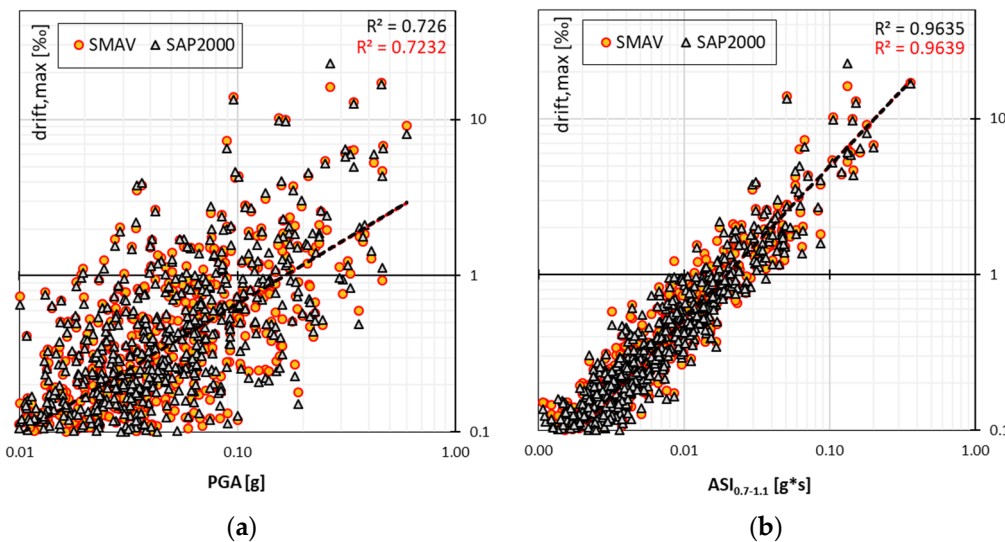

**Figure 9.** Detail of the Maximum Drift Ratios comparison in (**a**) PGA and (**b**) ASI scale.

As the plot shows, a better correlation is obtained if the results are arranged in $ASI_{0.7-1.1}$ scale (integration of Sa on the range of 0.7–1.1 s). This aspect is crucial in this study, since it can be considered as a better correlation to expected damage than the PGA scale. The results of the two models (SMAV and FEM) can be compared in terms of $R^2$. The values show a reliable response of the simplified model in comparison of the high accurate one.

Finally, the fragility curves were computed for the drift values of 1, 2 and 3‰ in PGA and ASI IM scales as the Figure 10 reports.

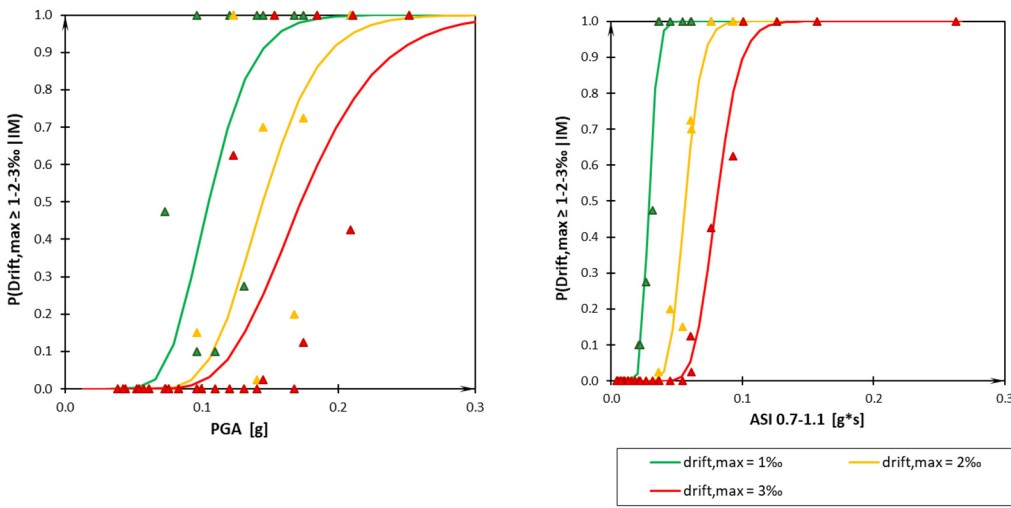

**Figure 10.** Fragility curves PGA and ASI of the SMAV model.

On the base of the results reported in Figure 9 the comparison between the before reported results of the SMAV model and the results obtained employing the nonlinear FEM model are reported in terms of ASI scale in Figure 11. In that figure, the curves are reported for 1‰, 3‰, 5‰ limits aiming to make some initial consideration on the differences between the two modelling approaches in non-linear field (5‰ limit).

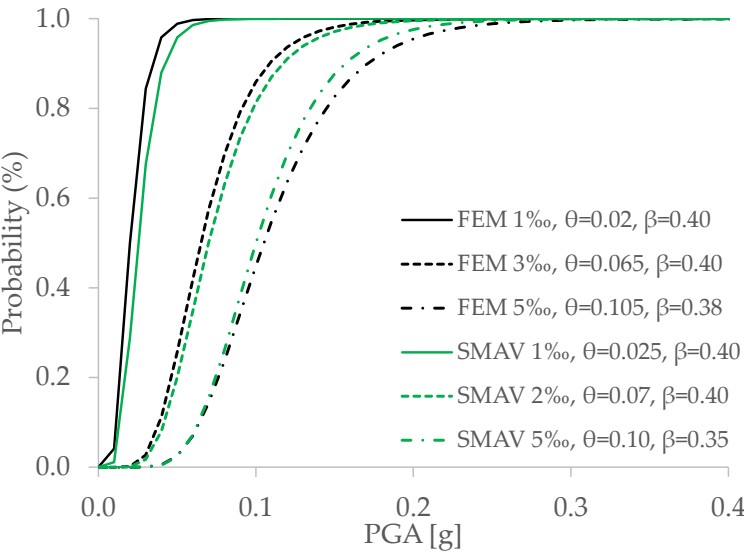

**Figure 11.** Fragility curves ASI comparison between FEM model and SMAV model.

Figure 11 compares the results in terms of PGA and ASI scale. For each drift level (1‰, 3‰, 5‰) a satisfactory agreement has been obtained even if, as previously underlined, the two models are extremely different in terms of sophistication. In case of 1‰ and 3‰ drift ratios, the SMAV-based curves are always lower than the FEM-based curves. This aspect is in agreement with the simplifications of the SMAV model in comparison to the FEM model. Due to the extremely lower computational burden of SMAV model, the above-mentioned differences look acceptable. The adopted methodology is suggested for light (D1: drift max = 1–2‰) and moderate (D2: drift max = 3 ‰) damage level only since the SMAV model, for the reason already specified in Section 2, is limited to seismic conditions that do not determine significant damage of the construction.

## 5. Application to Real Buildings

This section illustrates the applications of the proposed methodology to two reinforced concrete frame buildings:

Case study 1 (CS1): "Lorenzo dè Medici" School in Barberino del Mugello (FI);
Case study 2 (CS2): "San Giovanni Evangelista" Hospital in Tivoli (Rome).

Figure 12 reports two views of the two case studies (CS1 and CS2). CS1, is two storeys building with an irregular plan, CS2 is a nine-storeys building with a regular plan and a negligible interaction with surrounding structures. The two case studies can be considered representative of low- and high-rise existing structures, respectively.

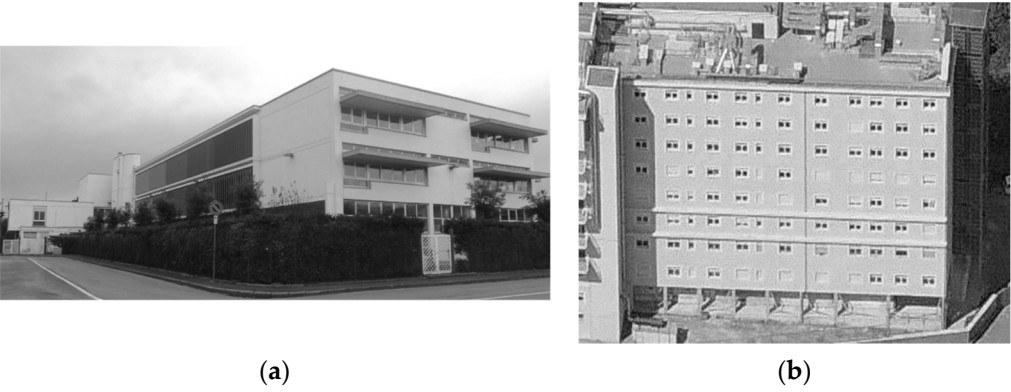

(**a**)                                          (**b**)

**Figure 12.** View of the two Case Studies (**a**) CS1 and (**b**) CS2 [39].

The SMAV models of the two buildings were implemented on the base of the experimental modal parameters provided by the Seismic Risk Service of the Civil Protection Department. The "Lorenzo dè Medici" School is also endowed with a permanent seismic monitoring system belonging to the Seismic Observatory of Structure of the Civil Protection Department [40].

Table 6 reports the modal properties of the CS1 (Case study 1). As the table shows, the first and second modes involve the structure mainly in one direction with a flexural behaviour. As matter of fact, the first mode activates only partially the structure as Modal Shape and My confirm. The third mode activates almost all the Mx mass and the structure act globally with a flexural behaviour.

**Table 6.** Modal properties of the case study CS1.

| Modes | Modal Shape | f [Hz] | $M_x$ | $M_y$ |
|:-----:|:-----------:|:------:|:-----:|:-----:|
| 1 | 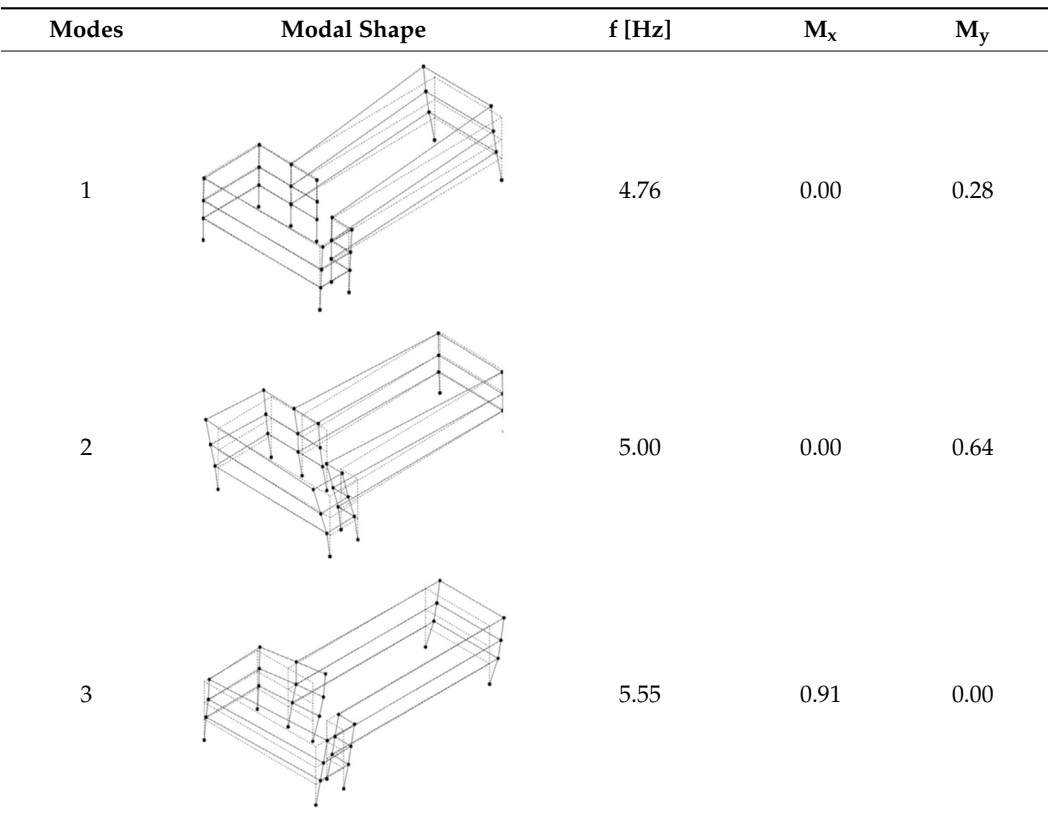 | 4.76 | 0.00 | 0.28 |
| 2 | | 5.00 | 0.00 | 0.64 |
| 3 | | 5.55 | 0.91 | 0.00 |

Table 7 summarises the modal behaviour of the CS2 building. As the table shows the first and second modes are flexural. Lastly the third mode is perfectly torsional. This behaviour is coherent with the building type.

As previously introduced, the structures have been investigated with a massive number of analyses. The results have been organised in Figures 13 and 14, respectively. The two figures report an image of the building and five plots. Each plot reports the IM scale versus the maximum drift. As matter of fact, the results are reported in terms of PGA, $Sa_{avg}$ and ASI scale. The regression is evaluated for each plot. Additionally, according to other studies [9] the ASI value was calculated for 0.1–0.5, 0.4–0.8 and 0.7–1.1 s. The three intervals can be considered representative of the first period values of low-, mid- and high-rise buildings.

**Table 7.** Modal properties of the case study CS2.

| Modes | Modal Shape | f [Hz] | Mx | My |
|---|---|---|---|---|
| 1 |  | 1.49 | 0.00 | 0.80 |
| 2 |  | 1.85 | 0.74 | 0.02 |
| 3 |  | 2.56 | 0.03 | 0.00 |

According to the results the following conclusions can be marked:

CS1: The parameter IM with the highest $R^2$ (0.8513) is ASI measured in the interval of periods 0.1–0.5 s, according to that expected (low-rise building);
CS2: The parameter IM with the highest $R^2$ (0.9492) is ASI measured in the interval of periods 0.7–1.1 s, according to that expected (high-rise building).

Additional it is worth to noting that:

(1) the highest correlation with $\delta_{MAX}$ is observed with the parameter ASI measured in the interval of periods of interest, therefore the ASI contributes better than the other parameters investigated in describing the earthquake damage potential;

(2) the $\delta_{MAX}$ and the PGA are weakly correlated, the higher the fundamental period of the building; this leads to discourage the construction of fragility curves in PGA especially for tall buildings;

(3) $\delta_{MAX}$ and $Sa_{avg}$ are well correlated for tall buildings, weakly correlated for low buildings; this is due to the different interval of integration periods. This aspect leads us to advise against the construction of fragility curves in $Sa_{avg}$ especially for low buildings. It can therefore be said that the best parameter for measuring the seismic intensity IM for the construction of fragility curves is the integral parameter ASI measured in the interval of the periods of interest.

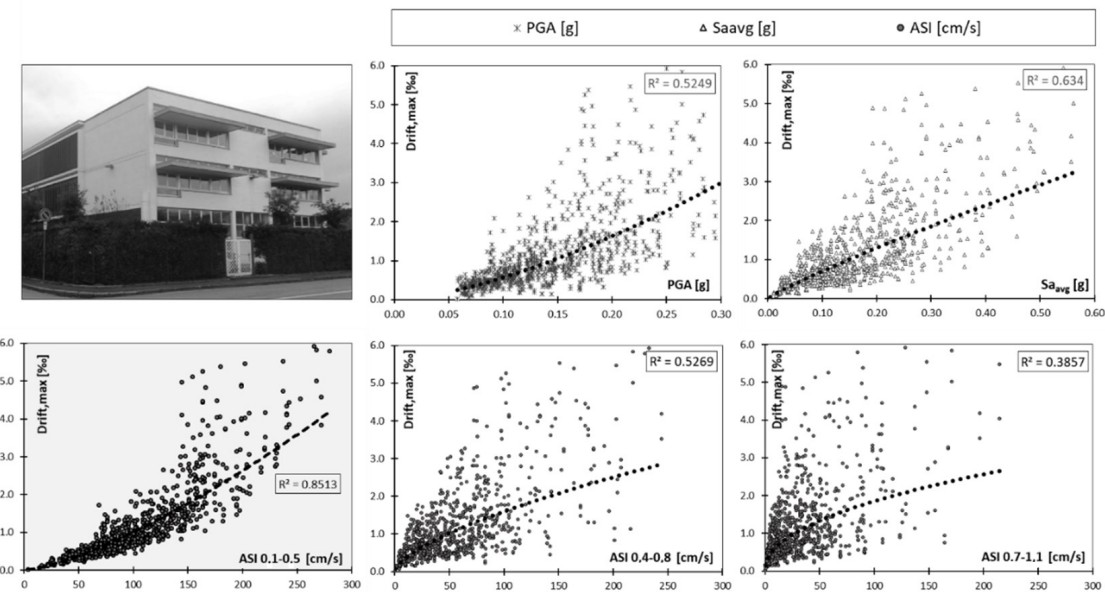

**Figure 13.** Results of the SMAV analysis of CS1.

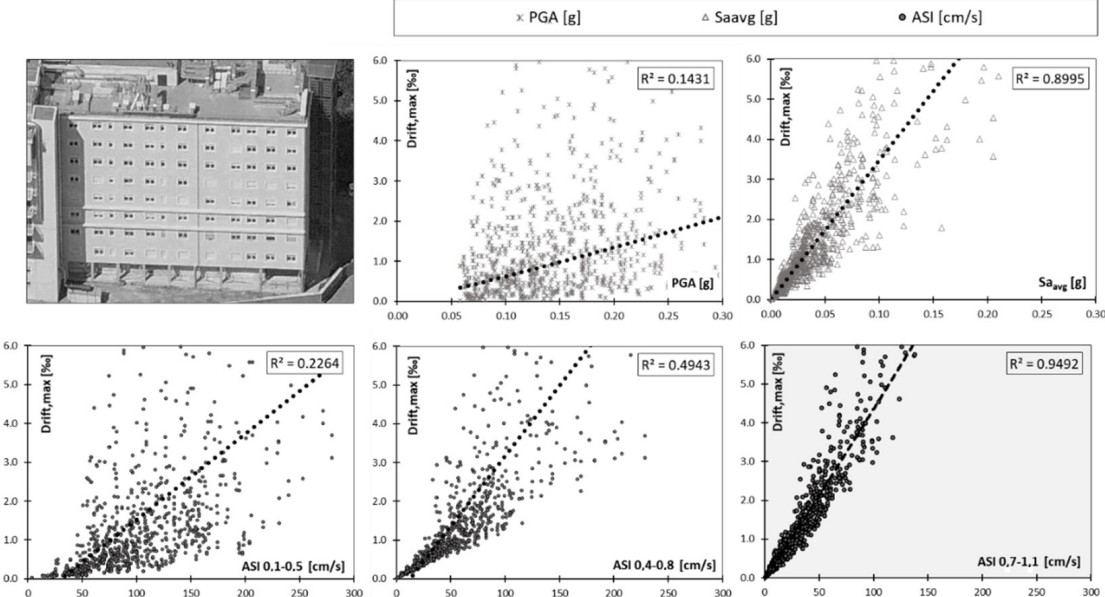

**Figure 14.** Results of the SMAV analysis of CS2.

However, it is worth to nothing that the three intervals are only a general interpretation of the first period of the structures. Even if other authors suggest to correlate the periods to the number of floors ([0.1, 0.5] s for buildings with 1, 2 or 3 floors, [0.4, 0.8] s for buildings with 4, 5 or 6 floors, [0.7, 1.1] s for buildings with more than 6 floors or base isolated [41–43]) an accurate experimental estimation of the first natural frequency (or period) is strongly suggested. This aspect is considered fundamental in terms of reliability of the result.

Finally, the fragility curves of the two case studies are reported in in terms of PGA (Figure 15) and in in ASI (Figure 16) scales.

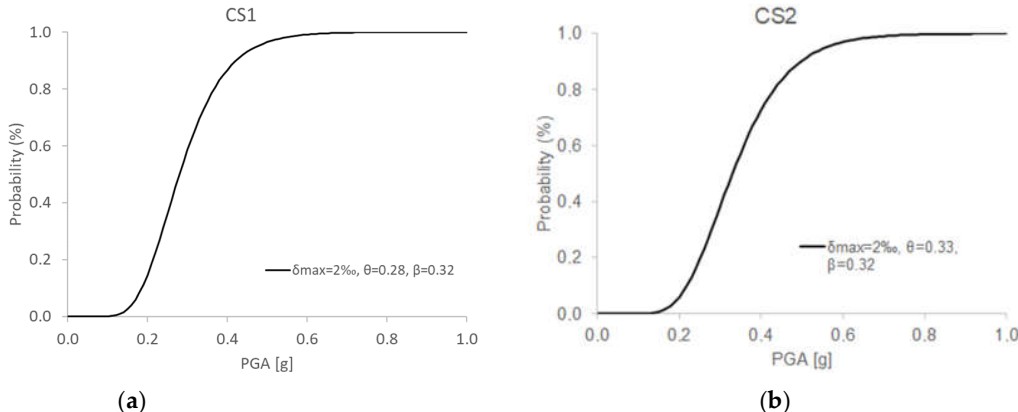

**Figure 15.** Fragility curves in PGA scale (**a**) CS1 and (**b**) CS2.

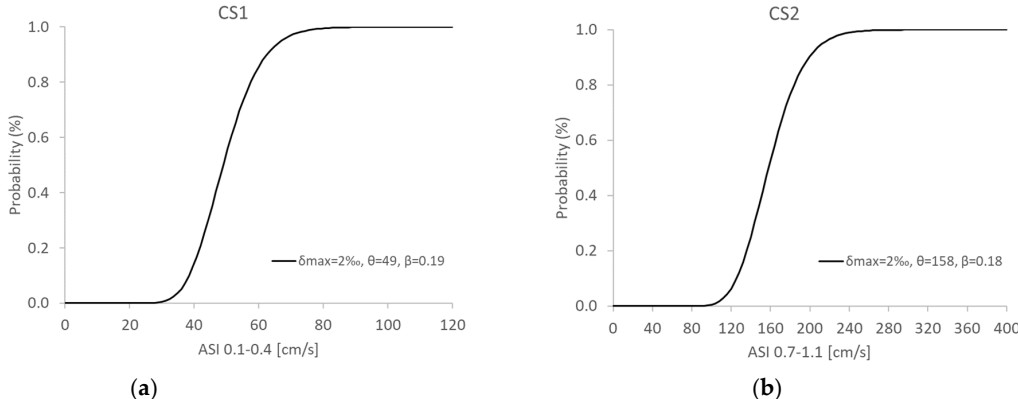

**Figure 16.** Fragility curves in ASI scale (**a**) CS1 and (**b**) CS2.

## 6. Conclusions

The amount of structures that were not designed to withstand earthquakes makes fundamental to establish reliable and low computationally demanding strategies for evaluating damage scenario at large or urban scale. This article presents a simplified SMAV-based framework for the evaluation of fragility curves applied to multi-storey buildings. The framework, starting from the modal properties obtained by dynamic identification campaigns, defines an SMAV model and through equivalent-linear response spectrum analyses proposes a procedure for the evaluation of the fragility curves. The proposed simplified computational framework is part of an ongoing research in collaboration with the Italian National Research Council and the seismic risk office of the Department of Civil Protection within the PON Governance 2014–2020 project on the reduction of seismic, volcanic and hydrogeological risk for civil protection purposes. Attempting to validate the framework, a numerical model that represents a ten-storey building, designed for vertical loads only, has been formulated with a nonlinear FEM approach and compared to the SMAV model in terms of fragility curves. The SMAV model results seem coherent with the more accurate nonlinear FEM method in the range of the considered interstorey drift limits even though, due the selected inputs database, the 3‰ and 5‰ drift threshold need additional investigations since the proposed procedure overestimates the expected drifts. Finally, the framework has been applied to two real cases and the fragility curves in PGA and ASI scales have been generated with an extremely computational burden reduction.

The main aspects that emerged from the application of the methodology to the case studies presented are summarized in the following points:

- the ASI integral parameter is better correlated with the $\delta_{max}$;

- the methodology allows to obtain fragility curves that mainly depend on the intrinsic characteristics of the building only if built in ASI.

Up to the moderate damage level, the fragility curves obtained with the SMAV model are very close to those obtained with the FEM model; therefore, SMAV can be adopted for fragility curves since the data necessary for its construction and its computational cost are more convenient than a FEM model.

The results encourage the authors to consider the framework as a very useful process for defining reliable building-specific fragility curves that can be adopted with economic and computational low burdens in view of seismic risk assessment process on several buildings. Additional improvements have to be implemented for considering the reliable of the results for high drift levels or interaction with surrounding buildings.

**Author Contributions:** Conceptualization, V.V. and D.S.; methodology, V.V. and G.O.; validation, V.V. and F.M.; formal analysis, V.V. and G.O.; writing—original draft preparation, G.O.; writing—review and editing, G.O. and D.S.; supervision, F.M. and D.S. All authors have read and agreed to the published version of the manuscript.

**Funding:** This research was funded by the Italian Department for Civil Protection (DCP) within the "PON Governance 2014–2020 project on the reduction of seismic, volcanic and hydrogeological risk for civil protection purposes—CIG 6980737E65—CUP J59G16000160006".

**Institutional Review Board Statement:** Not applicable.

**Informed Consent Statement:** Not applicable.

**Data Availability Statement:** Not applicable.

**Acknowledgments:** Authors want to thank Fabrizio Bramerini, Sergio Castenetto, Antonella Gorini, Giuseppe Naso (all of them from the Italian DCP) and Massimiliano Moscatelli (CNR-IGAG) for the useful discussions.

**Conflicts of Interest:** The authors declare no conflict of interest. The funders had no role in the design of the study; in the collection, analyses, or interpretation of data; in the writing of the manuscript, or in the decision to publish the results.

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
