# Peer review of "The Use of SMAV Model for Computing Fragility Curves"

_buildings, doi:10.3390/buildings12081213_

Round 1

Reviewer 1 Report

A very high quality research article has been prepared. The article is well written to academic standards, and the research works are robust. The work represents an important step towards a simpler method to evaluate safety, particular to those buildings not designed to modern seismic requirements. The reviewer agrees that the us of SMAV is an easy but accurate approach to verify structural models. With advancements of wireless sensors, measurements are now very easy.

line 247: please be specific which kind of nonlinearity is included? P-delta or material nonlinearity.

line 265: Here the frequencies from SAP2000 are reported but there is no mentioned that how the model was calibrated from ambient vibrations. 

Author Response

Reviewer #1:

The authors wish to thank the reviewer for his/her comments. In the following a detailed point-by-point response is reported.

A very high quality research article has been prepared. The article is well written to academic standards, and the research works are robust. The work represents an important step towards a simpler method to evaluate safety, particular to those buildings not designed to modern seismic requirements. The reviewer agrees that the use of SMAV is an easy but accurate approach to verify structural models. With advancements of wireless sensors, measurements are now very easy.

Comment #1 - line 247: please be specific which kind of nonlinearity is included? P-delta or material nonlinearity

RESPONSE: The text has been amended.

Comment #2 - line 265: Here the frequencies from SAP2000 are reported but there is no mentioned that how the model was calibrated from ambient vibrations.

RESPONSE: The building is a numerical benchmark only. It represents several buildings that were built in the south of Italy in the 70s but it is not an actual structure.

Reviewer 2 Report

The paper entitled “The use of SMAV model for computing fragility curves” provides a very interesting research work about the usage of Seismic Model from Ambient Vibrations procedures for the computation of the vulnerability of structures, in terms of fragility curves. The manuscript is well written, easy to read. However, a number of revisions are suggested, in order to increase the value of the manuscript.

1.      Literature review is good for all the covered topics. However, Authors are encouraged to provide some additional reference, (especially at Page 13 – Line 379), for experimental and numerical analyses of base isolated structures. Among the others, the following references are suggested:

Iervolino I., Baraschino R., Spillatura A. [2022] “Evolution of Seismic Reliability of Code-Conforming Italian Buildings”, Journal of Earthquake Engineering, DOI: https://doi.org/10.1080/13632469.2022.2087801

Cardone D., Viggiani L.R.S., Perrone G, Telesca A., Di Cesare A., Ponzo F.C., Ragni L., Micozzi F., Dall’Asta A., Furinghetti M., Pavese A [2022] “Modelling and Seismic Response Analysis of Existing Italian Residential RC Buildings Retrofitted by Seismic Isolation”, Journal of Earthquake Engineering, DOI: https://doi.org/10.1080/13632469.2022.2036271

O'Reilly G.J., Yasumoto H., Suzuki Y., Calvi G.M., Nakashima M. [2022] “Risk-based seismic design of base-isolated structures with single surface friction sliders”. Earth. Eng. and Str. Dyn., DOI: https://doi.org/10.1002/eqe.3668

2.      At Page 6 – Line 225, Authors describe the mechanical definition of the concrete and steel materials, which has been based on the common practice of that time and supported by the results of structural health monitoring campaigns. Have materials been considered as deterministic quantities? What about considering materials as random variables with certain p.d.f.? Further discussion should be provided.

3.      Page 8 – Line 267: Authors are encouraged to better discuss about how the selection of seismic events should be a proper representation of the seismic hazard. For example, an additional figure with all the response spectra overlapped could be useful, or maybe groups of spectra, related to certain ranges of PGA values.

4.      Page 7 – Line 245 the FEM model implemented in SAP2000 is described. Authors are suggested to better describe all the non-linear behaviors which are implemented, for both plastic hinges and infills, maybe by showing an additional figure with the adopted hysteretic rules. In addition, a non-linear push-over curve would be useful to provide the reader to the global information of the building behavior.

Author Response

Reviewer #2:

The paper entitled “The use of SMAV model for computing fragility curves” provides a very interesting research work about the usage of Seismic Model from Ambient Vibrations procedures for the computation of the vulnerability of structures, in terms of fragility curves. The manuscript is well written, easy to read. However, a number of revisions are suggested, in order to increase the value of the manuscript.

The authors wish to thank the reviewer for having appreciated the paper and for all the comments and suggestions aimed at improving the final version of the manuscript.

In the following a detailed response to each suggested point is reported.

Comment #1 Literature review is good for all the covered topics. However, Authors are encouraged to provide some additional reference, (especially at Page 13 – Line 379), for experimental and numerical analyses of base isolated structures. Among the others, the following references are suggested:

Iervolino I., Baraschino R., Spillatura A. [2022] “Evolution of Seismic Reliability of Code-Conforming Italian Buildings”, Journal of Earthquake Engineering, DOI: https://doi.org/10.1080/13632469.2022.2087801

Cardone D., Viggiani L.R.S., Perrone G, Telesca A., Di Cesare A., Ponzo F.C., Ragni L., Micozzi F., Dall’Asta A., Furinghetti M., Pavese A [2022] “Modelling and Seismic Response Analysis of Existing Italian Residential RC Buildings Retrofitted by Seismic Isolation”, Journal of Earthquake Engineering, DOI: https://doi.org/10.1080/13632469.2022.2036271

O'Reilly G.J., Yasumoto H., Suzuki Y., Calvi G.M., Nakashima M. [2022] “Risk-based seismic design of base-isolated structures with single surface friction sliders”. Earth. Eng. and Str. Dyn., DOI: https://doi.org/10.1002/eqe.3668

RESPONSE. The authors thank for the references that improve the manuscript

Comment #2 At Page 6 – Line 225, Authors describe the mechanical definition of the concrete and steel materials, which has been based on the common practice of that time and supported by the results of structural health monitoring campaigns. Have materials been considered as deterministic quantities? What about considering materials as random variables with certain p.d.f.? Further discussion should be provided.

RESPONSE. The authors thank the reviewer for the comment that underlines a key point in the assessment of the existing structures. Even the authors agree on the opportunity of a probabilistic approach on the material properties that can be defined in case of in-situ tests, this aspect has been considered out of the aim of the paper. The mechanical properties have been considered in a deterministic approach and the benchmark, that is not an actual building but represents a building typology, was considered for comparing the results of SMAV and FEM models. The probabilistic variation of the material properties and effects on the reliability of results may be investigated in future works.

Comment #3 Page 8 – Line 267: Authors are encouraged to better discuss about how the selection of seismic events should be a proper representation of the seismic hazard. For example, an additional figure with all the response spectra overlapped could be useful, or maybe groups of spectra, related to certain ranges of PGA values.

RESPONSE. Figure 6, of the original manuscript, already contains information about the PGA and ASI values of the adopted accelerograms. The horizontal axes represent PGA and ASI, respectively. The authors hope that the reviewer agree on the inopportunity of additional figures.

Comment #4 Page 7 – Line 245 the FEM model implemented in SAP2000 is described. Authors are suggested to better describe all the non-linear behaviors which are implemented, for both plastic hinges and infills, maybe by showing an additional figure with the adopted hysteretic rules. In addition, a non-linear push-over curve would be useful to provide the reader to the global information of the building behavior.

RESPONSE. The description has been improved by performing and commenting pushover analyses in both directions.

Reviewer 3 Report

General comments:
This is a very interesting study that explore the possibility of building fragility curves for building frames using only information on the building geometry and ambient vibration recordings of accelerometers placed at the various floors of the building.
The manuscript should be improved by clarifying how the SMAV approach is implemented in practice (particularly to estimate the nonlinear behaviour of the building) and by providing also more details on the SAP2000 model.
The English of the manuscript should also be carefully revised and improved.

Other comments:
Line 118-121. Please rewrite the sentence "It is important to.." and clarify what is the event corresponding to the probability of exceedance of 0.5.

line 147. Use italic for px, py and p.

Line 163. Define the Acceleration Spectrum Intensity, ????a−?b, and Ta and Tb.

Line 174. Why is the argument of P displayed as pedex?

line 179. Use "Multiple-Stripe Analysis" instead of "Multiple Stripes Analysis"

line 272-284. This section lacks details on how the two different models are constructed. Please add details.

line 284. Define the ASI 0.7-1.1 scales.

Author Response

Reviewer #3:

This is a very interesting study that explore the possibility of building fragility curves for building frames using only information on the building geometry and ambient vibration recordings of accelerometers placed at the various floors of the building.

The manuscript should be improved by clarifying how the SMAV approach is implemented in practice (particularly to estimate the nonlinear behaviour of the building) and by providing also more details on the SAP2000 model.

The English of the manuscript should also be carefully revised and improved.

The authors wish to thank the reviewer for appreciating the paper and for the suggested revisions.

Comment #1 Line 118-121. Please rewrite the sentence "It is important to.." and clarify what is the event corresponding to the probability of exceedance of 0.5.

RESPONSE. The authors thank the reviewer for the comment which gave them the opportunity to improve the text, specifying that the probabilistic character of SMAV is related not to the uncertainty on the seismic input, which is assumed to be known, but to the uncertainty on the mechanical and geometric characteristics of the structure.

Comment #2 line 147. Use italic for px, py and p.

RESPONSE. Amended

Comment #3 Line 163. Define the Acceleration Spectrum Intensity, ????a−?b, and Ta and Tb..

RESPONSE. Amended.

Comment #4 Line 174. Why is the argument of P displayed as pedex?

RESPONSE. Amended.

Comment #5 line 179. Use "Multiple-Stripe Analysis" instead of "Multiple Stripes Analysis"

RESPONSE. The term is coherent with the reference [24].

Comment #6  line 272-284. This section lacks details on how the two different models are constructed. Please add details.

RESPONSE. Amended.

Comment #6  line 284. Define the ASI 0.7-1.1 scales.

RESPONSE. Amended

Round 2

Reviewer 2 Report

Authors have carefully addressed all the Reviewers' comments and suggestions. Therefore, the manuscript can be accepted for publication in its revised form.

Author Response

The authors thank the reviewer.

Reviewer 3 Report

Most of my comments have been satisfactorily addressed.

The presentation of the SMAV method and of the "Frequency Shift Curves" (mentioned only in Section 2) must be improved. It is still not clear how they are applied.

The English of the paper should also be thoroughly revised and improved.

Author Response

ROUND 2

Comment #1  The presentation of the SMAV method and of the "FrequencyShift Curves" (mentioned only in Section 2) must be improved. Itis still not clear how they are applied.

RESPONSE. The mathematical model, used for simulating the seismic behaviour of the building, originates from SMAV (Seismic Model from Ambient Vibrations): a model developed by the authors aimed to predict the seismic response of masonry (Spina et al. 2018) and reinforced concrete buildings (Spina et al. 2021), based on the identification of the experimental modal parameters from ambient vibrations measurements. SMAV employs the Multi Rigid Polygons model (Acunzo et al. 2018) in order to estimate the seismic mass associated to each mode shape and for extrapolating the dynamic response of the building even in points not directly measured during the ambient tests. From an operative point of view, SMAV is obtained by ideally dividing each floor of the building into several polygons, which are assumed to have a rigid behaviour in their horizontal plan, and measuring ambient vibrations with two biaxial accelerometers placed on each of these polygons. Once natural frequencies and mode shapes have been extracted from the recorded signals, with one of the available operational modal analysis techniques, mode shapes are expressed in terms of rigid Degree of Freedom (DOFs) of each polygon (two translations and one rotation) with respect to their geometric center of gravity, in which the inertial properties of the building are also concentrated. In this way it is possible to compute the mass matrix of the building and so to obtain the seismic mass associated with each mode shape. The final result is a Predictive Modal Model of the building, in which each mode is described by natural frequency, damping ratio, mode shape and modal mass (and therefore seismic participation coefficient). The linear model thus obtained provides the seismic response of the building by mode superposition, first in the rigid DOFs of the polygons and then, through a simple linear transformation, in the DOFs of points along the perimeter of each floor of the structure.

In order to take into account the nonlinear behaviour of the building, SMAV is endowed with the so called “Frequency Shift Curves” developed for some specific structural typology, which allow the roof drift of the building during the earthquake to be correlated with the decrement (shifting) of its the apparent natural frequencies. Such curves have a probabilistic nature and therefore they do not provide crisp results but only value ranges.

Details can be found in reference [7,8].

The text has been improved.

Comment #2 The English of the paper should also be thoroughly revised and improved.

RESPONSE. The authors thank the reviewer and the text has been carefully checked.
